# Suspected Adenovirus Causing an Emerging HEPATITIS among Children below 10 Years: A Review

**DOI:** 10.3390/pathogens11070712

**Published:** 2022-06-21

**Authors:** Ali A. Rabaan, Muhammed A. Bakhrebah, Majed S. Nassar, Zuhair S. Natto, Abbas Al Mutair, Saad Alhumaid, Mohammed Aljeldah, Mohammed Garout, Wadha A. Alfouzan, Fatimah S. Alshahrani, Tarek Sulaiman, Meshal K. AlFonaisan, Mubarak Alfaresi, Saleh A. Alshamrani, Firzan Nainu, Shin Jie Yong, Om Prakash Choudhary, Naveed Ahmed

**Affiliations:** 1Molecular Diagnostic Laboratory, Johns Hopkins Aramco Healthcare, Dhahran 31311, Saudi Arabia; 2College of Medicine, Alfaisal University, Riyadh 11533, Saudi Arabia; 3Department of Public Health and Nutrition, The University of Haripur, Haripur 22610, Pakistan; 4Life Science and Environment Research Institute, King Abdulaziz City for Science and Technology (KACST), Riyadh 11442, Saudi Arabia; mbakhrbh@kacst.edu.sa (M.A.B.); mnassar@kacst.edu.sa (M.S.N.); 5Department of Dental Public Health, Faculty of Dentistry, King Abdulaziz University, Jeddah 21589, Saudi Arabia; znatto@kau.edu.sa; 6Research Center, Almoosa Specialist Hospital, Al-Ahsa 36342, Saudi Arabia; abbas.almutair@almoosahospital.com.sa; 7College of Nursing, Princess Norah Bint Abdulrahman University, Riyadh 11564, Saudi Arabia; 8School of Nursing, Wollongong University, Wollongong, NSW 2522, Australia; 9Nursing Department, Prince Sultan Military College of Health Sciences, Dhahran 33048, Saudi Arabia; 10Administration of Pharmaceutical Care, Al-Ahsa Health Cluster, Ministry of Health, Al-Ahsa 31982, Saudi Arabia; saalhumaid@moh.gov.sa; 11Department of Clinical Laboratory Sciences, College of Applied Medical Sciences, University of Hafr Al Batin, Hafr Al Batin 39831, Saudi Arabia; mmaljeldah@uhb.edu.sa; 12Department of Community Medicine and Health Care for Pilgrims, Faculty of Medicine, Umm Al-Qura University, Makkah 21955, Saudi Arabia; magarout@uqu.edu.sa; 13Department of Microbiology, Faculty of Medicine, Kuwait University, Safat 13110, Kuwait; alfouzan.w@ku.edu.kw; 14Microbiology Unit, Department of Laboratories, Farwania Hospital, Farwania 85000, Kuwait; 15Department of Internal Medicine, College of Medicine, King Saud University, Riyadh 11362, Saudi Arabia; falshahrani1@ksu.edu.sa; 16Division of Infectious Diseases, Department of Internal Medicine, College of Medicine, King Saud University Medical City, King Saud University, Riyadh 11451, Saudi Arabia; 17Infectious Diseases Section, Medical Specialties Department, King Fahad Medical City, Riyadh 12231, Saudi Arabia; dr.tarek.sulaiman@gmail.com; 18Basic Medical Sciences, Majmaah University, Majmaah 11952, Saudi Arabia; m.alfonaisan@mu.edu.sa; 19Department of Pathology and Laboratory Medicine, Sheikh Khalifa General Hospital, Umm Al Quwain P.O. Box 499, United Arab Emirates; mubarak.alfaresi@skgh.ae; 20Department of Pathology, College of Medicine, Mohammed Bin Rashid University of Medicine and Health Sciences, Dubai 505055, United Arab Emirates; 21Department of Clinical Laboratory Sciences, College of Applied Medical Sciences, Najran University, Najran 61441, Saudi Arabia; saalshamrani@nu.edu.sa; 22Department of Pharmacy, Faculty of Pharmacy, Hasanuddin University, Makassar 90245, Indonesia; firzannainu@unhas.ac.id; 23Department of Biological Sciences, School of Medical and Life Sciences, Sunway University, Subang Jaya 47500, Selangor, Malaysia; 16020778@imail.sunway.edu.my; 24Department of Veterinary Anatomy and Histology, College of Veterinary Sciences and Animal Husbandry, Central Agricultural University (I), Selesih, Aizawl 796 015, Mizoram, India; dr.om.choudhary@gmail.com; 25Department of Medical Microbiology and Parasitology, School of Medical Sciences, Universiti Sains Malaysia, Kubang Kerian 16150, Kelantan, Malaysia

**Keywords:** autoimmune hepatitis, acute hepatitis, liver inflammation, adverse effects, molecular mimicry, pathology

## Abstract

In October 2021, a case of acute hepatic failure without any known cause was identified in the United States of America. Upon further investigation, other children aged 1–6 years were reported to have the same liver failure, and some of them were positive for adenovirus 41 type F. On 21 April 2022, the Centers for Disease Control and Prevention (CDC) released an alert after 74 cases were identified in United Kingdom (UK) between 5 and 8 April in children below 10 years of age, some of whom were also found to be positive for SARS-CoV-2. All the patients showed symptoms such as vomiting, diarrhea, jaundice, and abdominal pain. The patients’ liver enzymes were remarkably increased. A total of 650 cases had been reported from 33 countries as of 27 May 2022, among which 222 cases were reported in the UK alone. No connection with SARS-CoV-2 or its vaccine has been found so far. However, the suspected cause is adenovirus, including its genomic variations, because its pathogenesis and laboratory investigations have been positively linked. Until further evidence emerges, hygiene precautions could be helpful to prevent its spread.

## 1. Introduction

In late 2021, in a hospital in Alabama, United States of America (US), five children between 1 and 6 years of age were admitted to hospital with severe liver failure, including acute liver failure [1]. None of these patients were positive for hepatitis A, B, or C virus, but they had positive reports of adenovirus [2]. Subsequently, after further investigation, four more cases were identified with the same etiology. All were positive for adenovirus 41 type F, which causes gastroenteritis in children [3]. In February 2022, an alert at state level was generated, which led to the identification of 10 cases with the same pathology by the World Health Organization (WHO) [4]. The Greater Glasgow and Clyde (GGC) Health Board of the Scottish National Health Service (NHS) notified Public Health Scotland (PHS), on 31 March 2022, that five children aged 3–5 years had presented to the Royal Hospital for Children in Glasgow with severe hepatitis of unknown etiology within a three-week period [5]. In Scotland, the average number of cases of hepatitis of unknown cause per year was previously less than four [6].

On 5 April 2022, the WHO stated that the number of cases increased to 74 within 3 days in the UK [7]. On 30 April, overall, more than 200 cases were identified, with 1 death and 17 liver transplants. All the patients were less than 10 years of age, and 10 children underwent liver transplant before 26 April [8]. According to the available information, 43 patients in England have since recovered [9]. An additional 12 cases have been reported in the US, 12 in Israel, and 1 in Japan [10,11,12]. The clinical condition usually starts with gastroenteritis-like symptoms and then progresses to jaundice [13]. This syndrome is similar to acute hepatitis, with a significant increase in liver enzymes. Patients’ levels of aspartate aminotransferase (AST) and alanine aminotransferase (ALT) rise to more than 500IU/L, and symptoms include abdominal pain, diarrhea, gastric problems, and jaundice. The relationship between COVID-19 and hepatitis is not yet clear, but, surprisingly, out of the first 74 patients, 20 were COVID-19-positive [14]. Adenovirus and SARS-CoV-2 were the most prevalent viruses detected in the cases, which were screened for a variety of infectious diseases [15]. Adenovirus was found in 75.5% of the cases in England and 50% of the cases in Scotland. The subtyping of 11 cases from the UK revealed that they were all type F41, which was the same subtype found in several of the reported US cases [16].

The WHO and CDC are assisting the countries that have reported cases for hepatitis of unknown etiology through ongoing investigations and the collection of these countries’ data [17]. Countries use their hepatitis networks and clinical organizations, such as the European Association for the Study of the Liver, the European Society of Clinical Microbiology and Infectious Diseases (ESCMID), and the European Society for Pediatrics Gastroenterology, Hepatology, and Nutrition, to disseminate all available information (ESPGHAN) [1]. The common viruses that cause infectious hepatitis were not found. The majority of the occurrences were reported in children under the age of five, who developed symptoms of gastroenteritis (nausea and diarrhea) before developing jaundice [18]. A few children above the age of ten were also examined as part of the investigation. Clinicians and scientists are investigating other factors that are probably associated, including another infection, such as COVID-19, or an environmental source, because this pattern of symptoms is unusual for adenovirus as shown in Table 1 [14].

A recent study by Cao et al. (2022) recommended that in order to minimize the physical stress and burden on the liver, bed rest is recommended. To meet the nutrition requirements of the body and to offer a high-carbohydrate, low-fat, moderate-protein diet, enteral nutrition should be used as the primary source of nourishment for infected patients. Calories, water, vitamins, and trace elements should be given intravenously to individuals with inadequate oral intake. It is necessary to address hypoalbuminemia. Laboratory markers, such as liver-function test results, electrolyte levels, acid-base balance, and coagulation indicators, as well as clinical circumstances, should be regularly checked [19].

Before reaching the clinical conditions for liver transplant, anti-inflammatory and detoxifying drugs can be used to treat liver injuries or jaundice, vitamin K and other anticoagulation therapies can be started to control the abnormal blood coagulation functions, a diet with sufficiently high protein, lactulose or probiotics can also be administered to prevent and treat hepatic encephalopathy, controlling the hypovolemia status can be adopted to prevent and treat hepatorenal syndrome, and side treatments for the prevention and treatment of secondary infections can also be started [19,20].

## 2. Epidemiological Summary

Early epidemiological analyses of patients from the UK based on traveling questionnaires failed to uncover a noteworthy common exposure (including food, medicines, or toxins) [17]. The toxicological testing of the specimens collected as part of the UK investigation is still underway. Although the Scottish inquiry found epidemiological ties in two pairs of cases, no other clusters have been discovered [21,22]. To date, the majority of cases in all reporting countries have had no substantial prior medical history [11]. To help determine the underlying etiology, detailed epidemiological and laboratory investigations of the cases are currently ongoing.

In several non-blood samples from the UK cases reviewed, other adenoviruses were discovered [17]. Although the data on testing in the EU/EEA are limited, adenovirus was found in 10 of the cases reported. The UK has documented statistical exceedance in the identification of various viruses in the population compared to positive testing in prior years, including a significant recent exceedance in adenovirus detections in fecal samples among children aged 1–4 years [17]. Table 2 shows the number of cases reported in different countries as on 27 May 2022.

Several countries in Europe and the US have reported cases of this cluster of illnesses [8]. Israel was recently the scene of a reported incident [9]. Hepatitis in its mildest form is common in young children of up to five years of age. At present, smaller numbers of reported cases have been noted around the world. However, as there are reported cases of hepatitis, clinicians should proceed with clinical investigations to diagnose liver inflammation or hepatitis of unknown etiology, and they should also examine the clinical condition of patients [24]. Table 3 shows the clinical picture of reported cases.

## 3. Risk Assessment of Disease

At present, the most strongly suspected causative agent for these lethal diseases is adenoviruses. Viral or immune pathological causes of liver damage may also be involved [25]. As a result of reduced exposure during the COVID-19 pandemic, prior SARS-CoV-2 or other infection, or a previously unknown coinfection or toxin, a normal adenovirus may induce a more severe clinical presentation in young children. The adenovirus 41 type F is primarily transmitted through the fecal–oral route and affects the gut. It is a common cause of pediatric acute gastroenteritis, which includes symptoms such as diarrhea, vomiting, and fever, as well as other respiratory symptoms. Alternatively, a new strain of adenovirus may have emerged with a different set of features [15].

It is only an assumption that adenovirus could be the reason behind this mysterious syndrome. The actual reason for this unexpected increase of acute hepatic failure cases, which have since been identified in several other countries, is yet to be confirmed [11,26]. Demographics, disease symptoms, medical and medication histories, family structure, recent household/close-contact illness, parental employment, food and water intake, health-service utilization, travel, animal exposure, and probable toxicant exposures were all investigated [27]. The first 60 patients with data were demographically similar to the overall hepatitis cohort, with no notable features or common exposures in terms of travel, family structure, parental occupation, diet, water source, animal exposure, or potential toxicant exposures, and no link to prior immunosuppression [28].

Usually, adenoviruses are contiguous, which causes respiratory-tract infections, and only few strains are involved in causing infections other than in the respiratory tract [2]. Type 41 adenovirus can cause hepatitis in immunocompromised children [3]. During the COVID-19 pandemic, the prevalence of adenovirus infections was significantly decreased, which might lead to an emergence of mutated novel adenovirus strains and might also be involved in adenovirus/COVID-19 coinfections [14]. Furthermore, because of the vaccination of young children against COVID-19, there could be a risk of the emergence of adenovirus/COVID-19 coinfections [29].

World healthcare monitoring bodies, such as the WHO and CDC, are continuously working to obtain clues regarding the pathogenic mechanism behind this phenomenon [1,22]. Adenovirus exceedances were also seen in routine laboratory data, primarily driven by enteric samples from the 1-to-4-year-old age group, although exceedances are also currently seen in many other common gastrointestinal and respiratory viral infections, most likely due to behavioral changes and population susceptibility following a period of low incidence during the pandemic [30]. Six patients were reported with positive Epstein–Barr virus (EBV) PCR test results but negative EBV immunoglobulin M (IgM) antibody test results (one patient did not undergo IgM testing), indicating that these were most likely not acute infections, but rather the low-level reactivation of previous infections [15].

Enterovirus/rhinovirus, meta-pneumovirus, respiratory syncytial virus, and human coronavirus OC43 were among the viruses found in some patients [31]. In a previous study, in total, 97 samples from patients and 75 healthy control samples from people of the same age were submitted for analysis [32]. The serum samples were analyzed for chemical substances, while the whole-blood and urine samples were analyzed for metals. These analyses took four different forms: polar and nonpolar, each with the positive and negative ionization of organic molecules and metabolites, which were targeted using liquid chromatography/high-resolution mass spectrometry (LC/HRMS). ICPMS was used for the metals, and GC/MS were employed for the analysis of volatile and semi-volatile organic compounds [32]. Qualitative reports of the organic masses were provided by the LC/HRMS, which were compared to various databases to identify substances. To identify possible chemicals of interest, the substances were compared to healthy controls and infected patients. If a reference standard is available, it can be used to corroborate the identification and quantitative assessment of potential acute hepatotoxic chemicals [25].

## 4. Hepatitis

The inflammation of the liver is referred to as hepatitis. The liver is an important organ that helps the body to digest nutrients, filter blood, and fight various infections [33]. The functions of the liver might be affected when it is inflamed or damaged. Hepatitis can be caused by excessive alcohol consumption, pollutants, certain drugs, and infections, especially those that are due to viruses and certain other medical disorders [34]. Despite having different routes of transmission, viruses can affect the liver initially and are known as viral liver infections. Hepatitis A, B, C, and D are common types of hepatitis. Patients with this kind of infection are also prone to cytomegalovirus (CMV) and EBV [15].

### 4.1. Pathophysiology of Viral Hepatitis

Viruses enter the blood stream through different transmission routes. When an adenovirus infects a person, it replicates in the epithelial cells of the lungs and other gastrointestinal organs. The virus begins to block host macromolecular synthesis and mRNA transport to the cytoplasm after many replication cycles. These cellular disruptions damage host cells and may produce symptoms of disease in their hosts, such as respiratory stress [35]. Studies have suggested that the penton protein of adenoviruses seems to be virulent, in addition to the symptoms caused by cell lysis and the inhibition of cellular synthesis. In laboratory settings, the penton protein causes cells to separate from monolayers. However, the significance of this discovery has not yet been determined in a clinical environment. Adenoviruses are known for their long latency. The virus may stay latent in the host even after the disease symptoms have faded [36].

The three stages of symptoms are prodromal, icteric, and convalescent. The symptoms differ depending on the clinical/infection phase of the patient [37]. When the virus enters the bloodstream, it starts releasing compounds throughout the prodromal stage. Fever, exhaustion, headache, vomiting, nausea, joint problems, and skin rashes are among the symptoms caused by these substances. Because the bile ducts and hepatocytes are damaged in the icteric stage, conjugated bilirubin and transaminases are released into the blood [16]. The patient appears yellow and produces dark urine because of the conjugated and unconjugated bilirubin. The symptoms start to improve and the patient returns to normal conditions during the recovery stage [11].

### 4.2. Severe Acute Hepatic Failure

In children, hepatitis is not uncommon, although the severity of previously reported infections is not too severe compared to the recently reported cases [38]. In young children, the known causes of hepatitis are still a potential source of the disease. Early diagnostic evidence raises the suspicion of adenovirus, but further evidence is needed before reaching a reasonable conclusion [2]. Adenoviruses have previously been linked to severe hepatitis in children, but there are other factors and infectious agents that have not yet been investigated [3].

### 4.3. Laboratory Assessment

At present, there are no specific diagnostic criteria available to diagnose this acute hepatitis of unknown origin. However, serum ALT and AST levels (>500 U/L), with or without elevated bilirubin levels could be evaluated in suspected cases (patients presenting indications of acute hepatitis), as well as initial screening tests for hepatitis A–E virus infections [1]. Elevated ammonia levels and prolonged prothrombin time on testing could also be used to determine the progression of disease. Lipid profiling, kidney function tests, complete stool analysis, routine urine analysis, and complete blood counts could also be very helpful in the diagnosis of severe acute hepatitis. Immunoglobulins, lymphocyte subsets, creatine kinase, cardiac enzymes, C-reactive protein, autoimmune hepatitis antibodies, anti-nuclear antibody, screening pf poisons, and heavy metals may be tested in accordance with the signs and symptoms [20,39].

Apart from these clinical laboratory examinations, the use of abdominal ultrasound could also be beneficial to rule out other diseases. If the radiological examination supports a clear diagnosis of severe acute hepatitis, nasal pharyngeal swabs, whole-blood samples, vomit samples, urine, and stool can be collected immediately for further etiological diagnosis purposes. If possible, a liver biopsy should be collected and submitted to histopathological examination [25,29].

## 5. Adenovirus

Adenoviruses are non-enveloped, icosahedral viruses of medium size (90–100 nm) that contain double-stranded DNA [3]. Human infections can be caused by more than 50 immunologically different adenovirus types. Common disinfectants are unable to kill adenoviruses, although they can be found on surfaces such as doorknobs, items, and even the water of swimming pools and small lakes. Adenoviruses usually infect the respiratory tract, eyes, urinary tract, intestines, and central nervous system [5]. Fever, sore throats, coughing, pink eye, and diarrhea are all frequent symptoms. Infections affect children more frequently than adults, but adenovirus can affect any age group [15].

These infections normally only cause minor symptoms and clear up on their own after a few days. They are, however, more dangerous in those with weakened immune systems, particularly young children [30]. Individuals with compromised immune systems are especially vulnerable to adenovirus infection, which can cause serious sickness. Some people with adenovirus infections, particularly those with weakened immune systems, can have undiagnosed infections in their tonsils, adenoids, and intestines [29]. They can spread their infection for extended periods of time. Certain individuals can shed the virus for weeks or months without showing any symptoms [30]. The diagnostic methods for adenovirus include antigen detection, polymerase chain reaction (PCR), virus isolation, and serology. Molecular approaches are typically used to identify adenoviruses [30].

### 5.1. Adenovirus and Hepatitis

Despite the fact that adenoviruses are prevalent diseases that normally result in self-limiting diseases [29]. They can, however, produce severe infections involving many body organs, including the liver, in the immunocompromised host. A review of Stanford University Medical Center’s pathology database from 1995 to 2016 found 12 cases of viral hepatitis, including biopsies and autopsy samples [11]. There were eight pediatric cases, seven of whom underwent orthotropic liver transplants and one of whom underwent lymphoblastic leukemia chemotherapy. There were four adult patients, one of whom underwent chemotherapy for chronic lymphocytic leukemia, while the other two received hematopoietic stem-cell transplants for malignant tumors. One child had lympho-plasmacytic cancer and underwent chemotherapy over one year before contracting adenovirus hepatitis, but he did not receive treatment at the time. Nonzonal coagulative hepatocyte necrosis and distinctive intra-nuclear inclusions were seen in all these cases. The extent of hepatocyte necrosis ranged from minor to severe. There was no concomitant inflammation in the majority of cases (7/12; 58%). If inflammation was detected, it was localized and lympho-histiocytic. The results were localized within the liver in one case, necessitating an image-guided biopsy. Adenovirus was not found in this patient’s non-targeted liver biopsy, which showed no histologic evidence of the virus. In the juvenile sample, 63 percent (5/8) of patients died from organ failure, but the adult population had a 100% (4/4) death rate [40].

In immunocompromised hosts, adenovirus hepatitis is a rapidly progressing and highly deadly infection [27]. Infections were particularly prevalent in patients who had received liver transplantation in the pediatric context. All of the adults, on the other hand, had hematologic malignancies and underwent chemotherapy or hematopoietic stem cell transplantation [10]. Coagulative necrosis and distinctive intra-nuclear inclusions are common histologic findings. Although this virus has been linked to severe hepatitis in immunocompromised individuals, it has not been linked to severe hepatitis in children with healthy immune systems. This is question remains unresolved [36].

### 5.2. Adenovirus 41 Type F

Type F41 of the adenovirus is a common cause of severe diarrhea and diarrhea-related death in young children across the world. It was first discovered in the stool of a child with gastroenteritis in the Netherlands in 1973 [41]. Type F41 has also been known to cause severe gastroenteritis in children; it is characterized by fever, vomiting, and diarrhea [42]. Adenovirus type F41 gastroenteritis may be life-threatening and is a common reason for hospitalization. The symptoms of acute gastroenteritis caused by adenovirus F41 are similar to those caused by other viruses. This intestinal adenovirus varies from respiratory and ocular adenoviruses in tissue tropism and pathogenicity, but the structural basis for this divergence remains unclear [43].

The findings reveal that adenovirus F41 often follows a varied endemic pattern of illness with rare outbreaks, suggesting that short-term cross-sectional, case-control, or cohort studies may be at danger of misrepresenting local incidence estimates [43]. The recent reported prevalence of adenovirus F41 infections in children with diarrhea was 13% in Guatemala [44], 5.1% in Nigeria [45], 1.5% in Brazil [46], 21.5% in Ethiopia [47], 56.9% in China [48], 28.5% in Thailand [49], and 62.5% in Iran [50]. The epidemiology and burden of adenovirus F41 remain poorly described because of the endemic pattern of the disease, with seasonal changes in incidence, and the requirement of molecular diagnostics to sensitively detect the organism [42].

## 6. SARS-CoV-2 and Hepatitis

A new clinical syndrome, known as COVID-19-induced hepatitis (CIH) has been discovered in SARS-CoV-2 patients and is defined as a benign new form of transient hepatitis, with characteristics such as gradual onset, elevated levels of AST and ALT, dilated sinusoids with lymphocytic infiltration of the liver parenchyma, non-obstructive jaundice, stable underlying liver disease, and no radiological change [14]. There is no evidence that COVID-19 has been involved in hepatitis infections among children [1]. However, it is not clear whether these two viruses are interacting. The intensity of this outbreak may be due to an enhanced vulnerability among young children that emerged during the COVID-19 pandemic, as a result of reduced adenovirus circulation in the past two years [51].

During the COVID-19 Omicron-variant wave in early 2022, the reported cases of COVID-19 infections in the US among children surged substantially, leading to 1,150,000 reported cases in one week. Nearly 88,000 COVID-19 cases among children were reported in the second week of June 2022 [52]. A recent study from the UK reported that between 28 December 2020 and 8 July 2021, a total of 109,626 UK children aged 5 to 17 years were proxy-reported. Of these children, a total of 60,050 (20,054 younger and 39,996 older) were tested for COVID-19 using PCR tests and 4,078 children were found positive [53]. However, a study from Israel from February 2020 to November 2021 reported that of the total hospitalization cases due to COVID-19 infections, 41% were children (5–11 years) [51].

Several respiratory viruses, including adenoviruses, persisted at lower levels during the COVID-19 pandemic, including adenoviruses in adults and children. A past or concurrent COVID-19 infection might influence the severity of adenovirus infections [20]. According to a recent study, 20 children (suspected adenovirus infected) have also tested positive for COVID-19 infection [24]. However, according to local officials, not all children were initially tested for COVID-19, and it was also unclear how many children had previously been infected with COVID-19. In fact, children with COVID-19 have been reported to have hepatic involvement. However, this involvement is usually accompanied by mild hepatitis and normal liver function. Rare incidences of severe hepatitis in children have been reported in the literature as part of COVID-19 or multiorgan dysfunction syndrome. The impact of concurrent or coincidental adenovirus/COVID-19 coinfections is still unknown [24].

To determine whether the combined effects of SARS-CoV-2 and adenovirus are more harmful to the liver, further research studies are needed. This may help, although the sensitivity and specificity of GADOUR criteria have not yet been demonstrated. Overly sensitive scoring systems require extensive statistical research before they can be established.

## 7. Conclusions

Currently, because the root cause of this disease is still unknown, it is suggested that suspected patients should be quarantined during diagnostic procedures and treatment. Objects contaminated by body fluids, feces, excrements, or blood should be decontaminated thoroughly. After the initial assessment of the patient, the case should be reported to a nearby health department. After obtaining relevant information about patients, based on signs and symptoms and laboratory reports, a multidisciplinary team comprising physicians from the departments of pediatrics, infectious diseases, emergency medicine, and intensive care should immediately begin further diagnosis and treatment for severe hepatitis of unknown origin. It is currently difficult to verify whether similar hepatitis cases have occurred in Europe, owing to the lack of comprehensive monitoring and study of hepatitis caused by human adenovirus infection. Furthermore, at present, it is difficult to conduct human adenovirus virological surveillance based on the clinical symptoms, and the potential risk of human-adenovirus-related hepatitis should be investigated as soon as possible using available relevant epidemiological, clinical, and virological data, as well as risk-factor information, to provide scientific and technical support for the prevention and control of this disease.

## Figures and Tables

**Table 1 pathogens-11-00712-t001:** The reported percentages of most common symptoms.

Symptoms	Percentage
Jaundice	74
Vomiting	73
Pale stool	58
Diarrhea	58
Fever	29
Nausea	39

**Table 2 pathogens-11-00712-t002:** Number of cases reported in different countries until 27 May 2022 [23].

Origin	Number of Cases
United Kingdom	222 (34.15%)
United States of America	216 (33.23%)
Japan	31 (4.76%)
Spain	29 (4.46%)
Italy	27 (4.15%)
Others	129 (19.84%)
Total	650

**Table 3 pathogens-11-00712-t003:** The clinical presentation of reported cases with the severity of disease [23].

Clinical Presentation	Frequency Strength	Disease Severity
Adenovirus positive	110 cases	18 had F41 type only
Age (<5 years)	490 cases	Hospital admissions in ICUs
SARS-CoV-2	23 cases	Co-infection
COVID-19 vaccination	10	-
Alanine amino-transaminase (ALT) > 500 IU/L	Majority	Severe jaundice
Liver transplant	38	Acute hepatic failure
Enteric tract involvement	Majority	Adenovirus F41 spread

## Data Availability

Not applicable.

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
