# Peer review of "Suspected Adenovirus Causing an Emerging HEPATITIS among Children below 10 Years: A Review"

_pathogens, 2022, doi:10.3390/pathogens11070712_

Round 1

Reviewer 1 Report

In this manuscript titled “Suspected Adeno virus causing an emerging Hepatitis among children below 10 years: A Review”, the authors reviewed the recent cases of hepatitis with unknown causes in young children. The authors nicely summarized the symptoms and epidemiology of these cases, and did a detailed description of the link between these cases and viruses, especially adenovirus. Overall, this is a timely and important review. Minor issues may need attention.

Minor issues:

1. It would be nice if authors could also summarize how these cases were managed besides liver transplantation as a last resort.

2. Section 6 SARS-CoV-2 and hepatitis lacks details, which is not very informative. For example, the authors mentioned “but surprisingly out first 74 patients 20 were COVID-19 positive [14]” in introduction (line 89), could the authors give some details in section 6 about how prevalent COVID-19 was in general young children population in those regions when those unknow hepatitis cases were reported? I would like to see if these hepatitis cases were enriched with COVID-19 infections or COVID-19 was just prevalent in these regions.

3. Line 155-156, “Notable point here is clinical presentation of adeno virus type 41 was not similar to current pathophysiology”, please elaborate.

4. Line 202-203, “As these viruses are intracellular pathogen host body’s cell mediated 202 immune response attacks the hepatocytes by killing them via natural killer T cells”, this is not accurate. NKT cells are not the only immune cell population which could kill virus infected hepatocytes.

5. Minor grammar issues should be noted, for example, line 160-162

6. Minor formatting issues should be noted, for example, line 131

Author Response

Reviewer 1

Comments and Suggestions for Authors

In this manuscript titled “Suspected Adeno virus causing an emerging Hepatitis among children below 10 years: A Review”, the authors reviewed the recent cases of hepatitis with unknown causes in young children. The authors nicely summarized the symptoms and epidemiology of these cases, and did a detailed description of the link between these cases and viruses, especially adenovirus. Overall, this is a timely and important review. Minor issues may need attention.

Minor issues:

  1. It would be nice if authors could also summarize how these cases were managed besides liver transplantation as a last resort.

Response: (Line 105-119) Two paragraphs has been added to summarize the general treatment possibilities for this disease.

  1. Section 6 SARS-CoV-2 and hepatitis lacks details, which is not very informative. For example, the authors mentioned “but surprisingly out first 74 patients 20 were COVID-19 positive [14]” in introduction (line 89), could the authors give some details in section 6 about how prevalent COVID-19 was in general young children population in those regions when those unknow hepatitis cases were reported? I would like to see if these hepatitis cases were enriched with COVID-19 infections or COVID-19 was just prevalent in these regions.

Response: (Line: 304-313) The literature about the prevalence of COVID-19 among children has been added in the revised manuscript.

  1. Line 155-156, “Notable point here is clinical presentation of adeno virus type 41 was not similar to current pathophysiology”, please elaborate.

Response: (Line: 169) The above-mentioned statement has been removed from this section as recommended by another reviewer, and we have added another separate subsection for adenovirus type 41.

  1. Line 202-203, “As these viruses are intracellular pathogen host body’s cell mediated 202 immune response attacks the hepatocytes by killing them via natural killer T cells”, this is not accurate. NKT cells are not the only immune cell population which could kill virus infected hepatocytes.

Response: (Line: 213-223) Section 4.1 has been replaced with the more updated literature and the previously written information has been removed as it was leading to misconception.

  1. Minor grammar issues should be noted, for example, line 160-162

Response: (Line: 160-165) The sentence has been revised.

  1. Minor formatting issues should be noted, for example, line 131

Response: (Line: 131-134) The sentence has been revised.

Reviewer 2 Report

1. In line 52, Flaviviridae family is not suitable to put here, as only HCV belongs to Flaviviridae, HAV, HBV, HDV, and HEV do not.

2. In line 54, it’s not in February 2022 but on April 21, 2022 that CDC issued a Health Alert Network (HAN) Health Advisory, which is different from a Health Alert that conveys the highest level of importance of the message.

3. In line 58, As of May 13, 2022, there have been 131 cases reported in the UK. There are more than 109 cases in the US, and in total there are more than 400 cases in 30 countries worldwide. Please check the data carefully and update the case number here and in Table 2. In table. 2, the case number was reported as of April 23, 2022 on the WHO website but not the date of May 6, 2022.

4. In line 70, please add reference here.

5. In line 73, the reference 4 was wrongly cited here, and please give reference to the claim of “February 2022, an alert at state level was generated…”.

6. In line 91, the etiology of these acute severe hepatitis in children has not confirmed as “infection”.

7. In line 99, “please give the context to the claim “there have been no deaths among the children”, or otherwise rephrase it.

8. In line 131, quotation mark should be not here.

9. In table 3, please include the updated data.

10. In line 229-230, “all the of patients, including the two transplant recipients, have healed or are recovering”, please either update the data or give the context to the patients.

11. In Adenovirus part, there is few details about Adenovirus 41 type F, but a lot of other adenovirus information, which is irrelevant to the hepatitis discussed in the manuscript.

12. The manuscript was not well organized or written, for instance, line 115-line 119 on page 3 should be considered the etiological assessment but not epidemiological results. Subsections 4.2 and 4.3 were not new and just repetitive to previous paragraphs. There were a lot of irrelevant adenovirus introduction but the useful information about AdV41 was missing.

13. Author contribution part was not finished.

14. An English proof-reading service is needed.

Author Response

Reviewer 2

Comments and Suggestions for Authors

  1. In line 52, Flaviviridae family is not suitable to put here, as only HCV belongs to Flaviviridae, HAV, HBV, HDV, and HEV do not.

Response: (Line: 52) The sentence has been corrected.

  1. In line 54, it’s not in February 2022 but on April 21, 2022 that CDC issued a Health Alert Network (HAN) Health Advisory, which is different from a Health Alert that conveys the highest level of importance of the message.

Response: (Line 54-56) The sentence has been revised and corrected.

  1. In line 58, As of May 13, 2022, there have been 131 cases reported in the UK. There are more than 109 cases in the US, and in total there are more than 400 cases in 30 countries worldwide. Please check the data carefully and update the case number here and in Table 2. In table. 2, the case number was reported as of April 23, 2022 on the WHO website but not the date of May 6, 2022.

Response: (Line: 58-59) The information about latest cases has been added in the revised manuscript. Also, table 2 has been amended accordingly.

  1. In line 70, please add reference here.

Response: (Line: 70) The reference has been added.

  1. In line 73, the reference 4 was wrongly cited here, and please give reference to the claim of “February 2022, an alert at state level was generated…”.

Response: (Line: 74) Reference 4 has been replaced with the correct reference.

  1. In line 91, the etiology of these acute severe hepatitis in children has not confirmed as “infection”.

Response: (Line: 91) The sentence has been revised and corrected.

  1. In line 99, “please give the context to the claim “there have been no deaths among the children”, or otherwise rephrase it.

Response: The context of the claim has been removed from the revised manuscript.

  1. In line 131, quotation mark should be not here.

Response: (Line: 131) The quotation mark has been removed.

  1. In table 3, please include the updated data.

Response: Data in table 3 has been updated.

  1. In line 229-230, “all the of patients, including the two transplant recipients, have healed or are recovering”, please either update the data or give the context to the patients.

Response: The laboratory assessment section has been totally revised with an updated literature. Furthermore, the above mentioned statement has been removed as it was causing misconception.

  1. In Adenovirus part, there is few details about Adenovirus 41 type F, but a lot of other adenovirus information, which is irrelevant to the hepatitis discussed in the manuscript.

Response: (Line: 308-324) Two new paragraph with separate subheading has been added in revised manuscript.

  1. The manuscript was not well organized or written, for instance, line 115-line 119 on page 3 should be considered the etiological assessment but not epidemiological results. Subsections 4.2 and 4.3 were not new and just repetitive to previous paragraphs. There were a lot of irrelevant adenovirus introduction but the useful information about AdV41 was missing.

Response: (Line: 242-258) The subsection 4.3 has been totally rewritten with most updated literature. Section 4.2. has been revised. (Line 115-119 has been shifted to Line 91-95). (Line: 308-324) Two new paragraph with separate subheading has been added in revised manuscript.

  1. Author contribution part was not finished.

Response: The author contribution part has been added.

  1. An English proof-reading service is needed.

Response: The manuscript has been thoroughly revised for English proofreading.

Reviewer 3 Report

The review “Suspected Adeno virus causing an emerging Hepatitis among children below 10 years: A Review” by Rabaan et al., covers a good understanding on the recent epidemiological studies. However, I believe that the writing parts can be improved and make more scientifically interesting. In addition, I have following minor comments for this paper.

1.     Line 67: Is it Alabama United States of America (US)? May be punctuation missing after Alabama? 

2.     Line 88: ‘also’ is not required.

3.     Line 89: should be out “of”

4.     Line 90: ECDC full name? 

5.     Please write the full sentence for the words from Line 100, 131, 142. For e.g., For We’re, aren’t, it’s….  Please correct it wherever required.

6.     F41 or 41F? should be consistent throughout the paper. See line 118 and in Table 3.

7.     Grammar: E.g., Line160-162. Please do not make a complex structure.

8.     Spelling of route: Line 200

9.     (Line 204) In chronic infections like Hepatitis B and C such process continuous for years may be [35].  It does not look a clear sentence.

10.  SARS-CoV2 and hepatitis part is very short. Please add more references and write more about this topic.

11.  Conclusion part can be improved.

Author Response

Reviewer 3

Comments and Suggestions for Authors

The review “Suspected Adeno virus causing an emerging Hepatitis among children below 10 years: A Review” by Rabaan et al., covers a good understanding on the recent epidemiological studies. However, I believe that the writing parts can be improved and make more scientifically interesting. In addition, I have following minor comments for this paper.

  1. Line 67: Is it Alabama United States of America (US)? May be punctuation missing after Alabama?

Response: (Line: 67) A punctuation between Alabama and US has been added in the revised manuscript.

  1. Line 88: ‘also’ is not required.

Response: (Line: 88) The word “also” has been removed.

  1. Line 89: should be out “of”

Response: (Line: 89) Corrected.

  1. Line 90: ECDC full name? 

Response: (Line 90) Apologies, as it was CDC. And the full abbreviation was provided at first appearace.

  1. Please write the full sentence for the words from Line 100, 131, 142. For e.g., For We’re, aren’t, it’s….  Please correct it wherever required.

Response: Corrected.

  1. F41 or 41F? should be consistent throughout the paper. See line 118 and in Table 3.

Response: Corrected.

  1. Grammar: E.g., Line160-162. Please do not make a complex structure.

Response: (Line: 160-165) The sentence has been revised.

  1. Spelling of route: Line 200

Response: (Line: 200) Corrected.

  1. (Line 204) In chronic infections like Hepatitis B and C such process continuous for years may be [35].  It does not look a clear sentence.

Response: (Line: 213-223) The paragraph has been replaced with more suitable literature.

  1. SARS-CoV2 and hepatitis part is very short. Please add more references and write more about this topic.

Response: (Line: 300-325) More literature about COVID-19 and hepatitis has been added in the revised manuscript.

  1. Conclusion part can be improved.

Response: (Line: 331-339) The conclusion section has been revised.

Round 2

Reviewer 2 Report

No major concerns.